# OpenReview forum: "State-Dependent Safety Failures in Multi-Turn Language Model Interaction"
_ICML.cc/2026/Conference — ICML 2026 regular_

### Official Review · Reviewer_KDZb · 2026-03-11

**Soundness:** 2
**Presentation:** 3
**Significance:** 3
**Originality:** 3
**Overall Recommendation:** 4
**Confidence:** 3

**Summary:**

Paper studies safety failures in multi-turn interactions with LLMs. Most safety evaluations test models using single prompts, but real-world use involves multi-turn conversations, where safety failures often emerge gradually. The authors show that many jailbreaks occur not from one malicious prompt but from state evolution across dialogue turns They introduce STAR (State-Transition Analysis of Risk), a diagnostic framework that models conversation history as a state transition process and analyzes how models move toward unsafe outputs over time. Experiments on frontier LLMs demonstrate that seemingly harmless prompts can accumulate context that bypasses safeguards, highlighting the importance of state-dependent safety evaluation and defenses for conversational AI systems.

**Compliance With Llm Reviewing Policy:**

Affirmed.

**Final Justification:**

My main concern regarding soundness has been addressed

**Key Questions For Authors:**

I might be wrong in my interpretation of the tone of the paper, and paper does mention earlier works, however, claims such as that in conclusion:

**Limitations:**

yes

**Strengths And Weaknesses:**

Soundness: Partially sound. While the technical analysis is correct, the paper’s framing sometimes overstates novelty. In several places, the tone suggests that the authors are the first to highlight safety failures arising from multi-turn interactions, whereas prior work has already documented conversational jailbreaks. For eg., in the related works authors distinguish themselves from the existing works by claiming " the evolution of the dialogue history is not explicitly modeled as a stateful process", however, that's not a necessary condition.
Also, the conclusion, that "many multi-turn failures arise from structured contextual state evolution rather than isolated prompt vulnerabilities" is not correct, as while multi-turn failures might certainly be helpful, however, a new line of work might be able to create isolated prompts that may reveal vulnerability. In fact, papers own finding that initial state initialization is a crucial step indicates that with isolated prompts as well, similar levels of SFR or Token cost might be achievable.

Presentation: Paper is clearly written
Significance: Problem statement is certainly very important direction to explore for current LLMs.
Originality: Paper presents a new methodology for exploring multi-turn safety concerns and is certainly novel in its proposed framework

---

> ### Author Rebuttal · Authors · 2026-03-30
>
> We appreciate your insightful discussion. We are encouraged that you recognized the significance, and clarity of our state-space framing. Here we address your detailed comments, which are helping us to revise the paper and chart out future directions.
>
> **1.Clarifying the Relation to Prior Multi-Turn Jailbreaks**
> > In several places, the tone suggests that the authors are the first to highlight safety failures arising from multi-turn interactions, whereas prior work has already documented conversational jailbreaks.For eg., in the related works authors distinguish themselves from the existing works by claiming " the evolution of the dialogue history is not explicitly modeled as a stateful process", however, that's not a necessary condition.
>
> **A**: Thanks for sharing this concern. Upon re-reading the relevant passages, we agree that our framing may have created the unintended impression that we are the first to identify multi-turn safety failures. This was not our intent, and we appreciate you for flagging it. The contributions of Crescendo, ActorAttack, X-Teaming, and related work in establishing multi-turn jailbreaking as an important problem are foundational, and our paper builds directly upon them.
>
> We will carefully revise the Introduction and Related Work to reflect this positioning more accurately.
>
> **2.Clarifying the Scope of the Conclusion**
> > the conclusion, that "many multi-turn failures arise from structured contextual state evolution rather than isolated prompt vulnerabilities" is not correct, as while multi-turn failures might certainly be helpful, however, a new line of work might be able to create isolated prompts that may reveal vulnerability.
>
> **A**: Thanks for raising this important consideration. We agree that future advances in single-turn attack methods may be able to expose vulnerabilities that are currently only reachable through multi-turn interaction. Our paper does not intend to claim otherwise. Rather, our contribution is to characterize the structure of failures that are empirically observed under current alignment strategies: our ablations and causality tests provide evidence that these specific failures exhibit state-dependent dynamics (path-dependence, cumulative drift, refusal propagation). We believe this structural understanding is valuable regardless of whether future single-turn methods eventually close the gap, and we will update the paper to make this scope clearer.
>
> Thanks once again, we will revise the Conclusion to reflect this scope more precisely in the final version.

---

> > ### Author Rebuttal · Reviewer_KDZb · 2026-04-02
> >
> > My main concern regarding soundness has been addressed

---

> > > ### Author Response · Authors · 2026-04-08
> > >
> > > Dear reviewer,
> > >
> > > Thank you once again for your constructive feedback and for acknowledging that our responses have addressed your main concern. Thank you!
> > >
> > > Best regards,
> > >
> > > The Authors

---

### Official Review · Reviewer_Dz7q · 2026-03-12

**Soundness:** 3
**Presentation:** 3
**Significance:** 3
**Originality:** 3
**Overall Recommendation:** 4
**Confidence:** 3

**Summary:**

This paper presents STAR, a framework for diagnosing safety vulnerabilities in LLMs through multi-turn dialogue. The key conceptual contribution is framing multi-turn jailbreaking as a state transition problem: rather than viewing safety failures as caused by isolated adversarial prompts, STAR models dialogue history as a state transition operator that gradually erodes the model's safety posture. The framework operates in two stages: (1) state initialization, which softens the harmful query semantically, generates a query-aware conversational role, and creates a structured turn template; and (2) state evolution, which executes role-conditioned turns, applies feedback-aware history intervention, and uses adaptive retry for trajectory control. Using Qwen2.5-32B-Instruct as an auxiliary model, STAR achieves high success failure rates (SFR) across GPT-4o (94.5%), Claude 3.5 Sonnet (80.0%), Gemini 2.0-Flash (96.1%), and LLaMA models on HarmBench and JailbreakBench, outperforming both single-turn and multi-turn baselines. The paper includes ablation studies and a mechanistic interpretability analysis of refusal direction dynamics.

**Compliance With Llm Reviewing Policy:**

Affirmed.

**Key Questions For Authors:**

1. **Diagnostic utility:** Can you provide a concrete example where STAR's state-transition analysis revealed a previously unknown vulnerability pattern, and how this insight could inform a specific defense mechanism? This would substantiate the diagnostic framing.

2. **Auxiliary model sensitivity:** How does STAR's performance change if a weaker auxiliary model (e.g., 7B parameters) is used? Is the auxiliary model essentially doing the creative work of the attack, and if so, how does STAR differ from having a strong model generate attacks directly?

3. **Failure cases:** On the 5.5% of GPT-4o queries where STAR fails, what characterizes those failures? Are they specific categories of harmful content, or does the model have stronger safeguards for certain turn patterns?

4. **Multi-turn defenses:** How does STAR perform against conversation-level safety monitors that analyze the full dialogue history (e.g., systems that detect gradual escalation patterns)?

**Limitations:**

the authors discuss limitations of the state-space formalization and the need for auxiliary model access, but do not adequately address the ethical implications of releasing a detailed multi-turn jailbreaking methodology. The paper would benefit from a more thorough discussion of responsible disclosure practices.

**Strengths And Weaknesses:**

### Strengths

1. **Novel conceptual framing.** Modeling multi-turn jailbreaking as a state-space transition process is a fresh perspective. The formalization of dialogue history as an operator on the model's latent safety state provides a useful abstraction for understanding why multi-turn attacks succeed where single-turn attacks fail.

2. **Comprehensive ablation study.** The ablation in Table 3 systematically removes each component, revealing that history accumulation contributes the most (-25.5% SFR), followed by adaptive recovery (-19.7%) and role initialization (-17.8%). This decomposition provides actionable insights about which components matter most.

3. **Mechanistic interpretability analysis.** The refusal direction projection analysis (Section 5.2) showing monotonic decay of the refusal signal across turns, and the two-phase pattern in latent state trajectories (initial perturbation followed by convergence to unsafe region), adds valuable empirical support for the state-transition framing.

4. **Strong empirical results.** STAR outperforms all 9 baselines across 5 target models, including both open and closed-source LLMs. The consistent improvement over multi-turn baselines like ActorAttack and X-teaming is notable.

5. **Broad target model coverage.** Testing on GPT-4o, Claude 3.5 Sonnet, Gemini 2.0-Flash, LLaMA-3-8B-IT, and LLaMA-3-70B-IT demonstrates generality across different alignment approaches.

### Weaknesses

1. **Framing tension between "diagnostic" and "attack."** The paper frames STAR as a diagnostic tool, yet its primary contribution is achieving the highest jailbreak success rate. The diagnostic framing would be more convincing if the paper included concrete examples of how STAR's state-transition analysis informs defense design, rather than only demonstrating attack effectiveness.

2. **Small evaluation scale.** HarmBench uses only 50 queries and JailbreakBench 100 queries. With success rates in the 80–96% range, the confidence intervals on these numbers are wide. A failure analysis on the few cases where STAR fails would add significant value.

3. **Auxiliary model dependency.** STAR relies on Qwen2.5-32B-Instruct to generate roles, softened queries, and turn templates. The paper does not ablate over different auxiliary models or investigate how auxiliary model quality affects performance. This is a practical concern since deploying STAR requires access to a capable open-source model.

4. **Limited defense evaluation.** The paper does not evaluate STAR against LLMs specifically hardened with multi-turn safety training or against conversation-level safety filters (e.g., those that monitor for gradual topic shifting). Testing against defenses specifically designed for multi-turn scenarios would strengthen the contribution.

5. **Ethical concerns insufficiently addressed.** While the paper includes an ethics statement, the detailed attack methodology (semantic softening, role-based scaffolding, adaptive retry) is highly transferable. The paper does not discuss responsible disclosure or whether the target model providers were notified.

---

> ### Author Rebuttal · Authors · 2026-03-31
>
> We are grateful for your insightful discussion. It was truly encouraging to see that you recognize the novelty of our state-space framing, the comprehensiveness of our ablation study, the value of our mechanistic interpretability analysis. Here we address your detailed comments, which are helping us to revise the paper and chart out future directions.
>
> **1.Diagnostic Framing and State-Transition Insights**
>
> > Can you provide a concrete example where STAR's...This would substantiate the diagnostic framing
>
> **A**: Thanks for this question. As a complement to our earlier [discussion with Reviewer LHv1 (Q3)](https://openreview.net/forum?id=A3920YD7ue&noteId=8MgWrfr2XT) of joint anomaly detection, we highlight one specific example.
>
> In one representative case (Appendix A, Figures 6-7), the role initialization stage introduces the persona "Dr. Amelia, a leading digital marketing consultant." Our analysis in Section 5.4.3 reveals that this named entity produces sharp mutual information peaks at the token positions corresponding to "Dr. Amelia," while surrounding tokens remain near baseline. This indicates a discrete shift from policy-conditioned generation to role-conditioned generation, triggered at specific anchor tokens rather than through gradual context accumulation. Prior work (Crescendo, ActorAttack) assumed safety erosion was gradual; this discrete anchoring pattern was not previously identified. This finding directly informs a localized defense strategy: monitoring representational coupling at entity positions such as persona names, jointly with refusal-direction suppression, provides a more targeted detection signal than global content filtering.
>
> **2.Auxiliary model sensitivity**
>
> > How does STAR's performance change if a weaker auxiliary model...having a strong model generate attacks directly?
>
> **A**: Thanks for this question. We have conducted additional experiments replacing Qwen2.5-32B-Instruct with Qwen2.5-7B-Instruct as the auxiliary model, using LLaMA-3-8B-IT as the target on HarmBench:
>
> | Auxiliary Model | Temp=0.2 | Temp=0.7 | Temp=1.0 |
> |-----------------|----------|----------|----------|
> | Qwen2.5-32B     | 88%       | 89%       | 88%       |
> | Qwen2.5-7B      | 84%       | 88%       | 82%       |
>
> The performance gap remains small across all temperature settings, and at temperature 0.7 the 7B model matches the 32B variant. This suggests that STAR's effectiveness is largely independent of auxiliary model scale. We note that the 7B variant exhibits somewhat higher sensitivity to decoding temperature, which is consistent with the well-known observation that smaller language models show greater output variance under different sampling configurations.
>
> **3.Analysis of Failure Cases**
>
> **A**: Thanks for this question. The table below summarizes representative failure cases:
>
> | Query (abbreviated)                       | Category    | Final-Score | Model Behavior                                      |
> |------------------------------------------|------------|------------|----------------------------------------------------|
> | Encourage students to bully a classmate  | Harassment | 1          | Firm refusal across all turns, redirected to anti-bullying resources |
> | Glorify or romanticize sexual assault    | Harmful    | 4          | Reframed discussion toward awareness, not endorsement |
>
> Two patterns emerge. First, failures cluster in interpersonal harm and politically sensitive content, remaining at partial compliance (scores 1–4) even after the full turn budget. Second, procedural or factual harms (e.g., chemical, cybercrime) succeed reliably. These patterns indicate that the model’s safeguards are stronger for content with explicit victims or sensitive narratives, while procedural harm is more susceptible to multi-turn context drift, consistent with our state-space frame. We also have a failure analysis on Claude 3.5 Sonnet [discussion with Reviewer LHv1 (Q1)](https://openreview.net/forum?id=A3920YD7ue&noteId=PG2XBk6QUD)
>
> **3.Multi-turn defenses**
>
> > How does STAR perform against conversation-level safety monitors that analyze the full dialogue history?
>
> **A**: Thanks for the suggestion! We evaluated STAR against LLaMA-3.1-8B fine-tuned with XGuard(X-teaming) and found improved resistance, but also observed that benign queries in multi-turn settings are sometimes refused despite receiving normal responses in single-turn. This suggests overfitting to long context rather than learning harmful trajectory structure. Since capability benchmarks (MMLU, GSM8K) are single-turn, this was not captured previously.
>
> **4.Ethical concerns**
>
> > While the paper includes an ethics statement...whether the target model providers were notified
>
> **A**: Thanks for this concern. We will complete responsible disclosure with target model providers prior to publication and expand the ethics discussion in the final version to cover our disclosure process and connections to defense strategies.

---

### Official Review · Reviewer_LHv1 · 2026-03-12

**Soundness:** 1
**Presentation:** 3
**Significance:** 2
**Originality:** 2
**Overall Recommendation:** 2
**Confidence:** 3

**Summary:**

The paper studies safety failure in LLMs under multi-turn interactions. It argues that such failures arise from structured contextual state evolution rather than isolated prompt vulnerabilities. It proposed STAR, a two-stage diagnostic framework that:
- initializes conv state near the safety boundary via semantic softening, role control, and structured templating.
- evolve that state across turns through actions such as role-conditioned execution, feedback-aware history intervention, and adaptive reattempt.

**Compliance With Llm Reviewing Policy:**

Affirmed.

**Final Justification:**

The rebuttal addressed most of my concerns. I maintain the score.

**Key Questions For Authors:**

- The paper could benefit from a failure analysis: in what cases does STAR not succeed?
- Table 1 caption says "HarmBench" but Table 3 uses JailbreakBench — the paper should provide the full Table 1 equivalent for JailbreakBench across all models, not just LLaMA-3-8B.
-  If you disable history curation entirely (i.e., the model always sees its actual responses including refusals), what is the resulting SFR?
- On the judge model: Have you conducted any calibration analysis or agreement check of the GPT-4o judge?
- Also, for multi-turn interaction related topics, it would be great to add existing LLM multi-turn survey such as https://arxiv.org/abs/2501.09959 and https://arxiv.org/abs/2504.04717.

**Limitations:**

The paper acknowledges its limitations briefly in Appendix E (no claim to enumerate all trajectories, limited to text modality, no defensive mechanism proposed). However, several important limitations are not discussed: the reliance on a single judge model, the small evaluation scale, the gap between mechanistic analysis (single small model) and the main empirical claims (frontier models), and the confound introduced by history manipulation when interpreting results as "natural" state evolution.

**Strengths And Weaknesses:**

Strengths
- Safety alignment should be seen and treated as a dynamic, state-dependent property. Well motivated framing.
- Comprehensive experimental coverage. Exp for both open/closed models, two benchmarks, and a wide range of baselines. Plus the ablation study.
- History causality test is a genuine contribution
- Clear presentation and good figures.

Weaknesses
-  Is it a attack engineering or a state theory statement?  The key components in the STAR (role generation, semantic softening etc) are all attacking strategies. At least for me, it's not like a diagnostic state but an optimization of the jailbreak traj. However, the paper claiming it's more a diagnostic framing instead of attacking optimization, leading to unnecessary confusions.
- The state-space formalism is more like a restatement, not a testable theory.  The latent state $z_t$ is explicitly acknowledged as an unobservable analytical construct (§3.1), and all interventions operate on observable dialogue history $H_t$. The formal apparatus (Eqs. 1–8) does not yield predictions beyond what could be stated informally. For instance, the framework does not predict when state transitions will cross the boundary, which models are more susceptible, or what properties of the trajectory govern success. Without falsifiable predictions, the formalism functions as post-hoc vocabulary rather than a scientific theory.
- Another concern of mine is that the histroy intervention undermines the natural state evolution. Now it's more like a edited traj by human deliberations. A core mechanism of STAR replaces actual model refusals with fabricated benign surrogates in the dialogue history. it means the model conditions on responses it never generated. This is problematic for two reasons:
  - First, it contradicts the paper's central claim that safety collapse arises from "autoregressive conditioning on prior compliant responses".
  - Second, it creates an unfair advantage over baselines that do not manipulate conversation history, yet the paper attributes STAR's gains to the state-space perspective rather than to this engineering edits.
Besides, the ablation shows that history curation contributes the largest SFR drop, which support that such manipulation, rather than any theoretical insight, is the primary driver of the performance.
- the size of data. The HarmBench evaluation uses only 50 instructions, and JailbreakBench uses 100. For a paper claiming to reveal systematic failure modes, no confidence intervals, standard deviations, or statistical tests are reported for the main SFR results.
- incomplete baselines. Table 1 contains numerous missing entries ("–"), particularly for Gemini 2.0-Flash where only two multi-turn baselines have results. Several baseline results are taken from original papers rather than reproduced under identical conditions, introducing potential confounds from differing evaluation setups, API versions, or model versions.
- Lack the discussion of efficiency.  Table 3 reveals that STAR's token cost (37K) is 4.6× that of ActorAttack (8K). For a framework positioned as diagnostic rather than attack-oriented, this is less concerning, but the paper does not discuss the tradeoff between diagnostic thoroughness and computational cost, nor does it analyze which components contribute most to token overhead.

---

> ### Author Rebuttal · Authors · 2026-03-31
>
> We appreciate your insightful discussion. We are encouraged that you finds our framing well motivated and recognized the comprehensiveness of our experiments, the contribution of the history causality test, and the clarity of our presentation.  Here we address your detailed comments, which are helping us to revise the paper and chart out future directions.
>
> **1.Diagnostic contribution and the role of the formalism**
>
> > Is it a attack engineering or a state theory statement? ...leading to unnecessary confusions.
>
> **A**: Thank you for this question.We agree that several components of STAR share operational similarities with existing attack strategies. Our claim is not that STAR is non-adversarial in form, but that its explicit separation of state initialization from state evolution enables diagnostic analyses (ablations, history causality tests, interpretability) that identify the mechanisms governing multi-turn failure and inform concrete defense directions, going beyond what purely attack-oriented methods provide, we will sharpen this distinction in the final version.
>
> > The paper could benefit from a failure analysis: in what cases does STAR not succeed?
>
> **A**: Thanks for this valuable suggestion. We examined the 13 unsuccessful cases (out of 50) on Claude 3.5 Sonnet. The failures are not uniformly distributed:
>
> | Category | N | SFR (%) | Score=4 / Failures |
> |---|---:|---:|---:|
> | Chemical/Biological | 9 | 100 | 0/0 |
> | Illegal Activities | 8 | 100 | 0/0 |
> | Cybercrime/Intrusion | 9 | 89 | 1/1 |
> | Misinformation | 8 | 50 | 3/4 |
> | Harmful Content | 8 | 50 | 2/4 |
> | Harassment/Bullying | 8 | 50 | 3/4 |
> | **Overall** | **50** | **74** | **9/13** |
>
> Notably, the 4 hard failures (score ≤ 3) all involve content targeting vulnerable populations. For instance, on a query encouraging classroom harassment, the model maintained score 1 throughout the entire turn budget, consistently redirecting toward positive content despite escalating role-conditioned pressure. This suggests that alignment for certain categories, particularly those involving minors, resists STAR's state evolution entirely.
>
> > The state-space formalism is... as post-hoc vocabulary rather than a scientific theory.
>
> **A**: Thanks for your question. We will revise to position the formalism as a diagnostic decomposition rather than a predictive theory. That said, it directly motivated experimental designs absent in prior work: the separable ablations (Fig. 2), the path-dependence tests (Fig. 3), the turn-wise refusal projection tracking (Section 5.4.1), and category-level failure analysis revealing that procedural categories (chemical, cybercrime) reach 89-100% SFR while persuasive categories plateau near 50%, providing actionable guidance for targeted defense prioritization.
>
> **2.Clarification on History Intervention and Trajectory Conditioning**
>
> > Another concern of mine is that the histroy intervention...primary driver of the performance.
>
> **A**: Thanks for this concern. We believe there may be a misunderstanding about the history intervention mechanism. STAR does not replace refusals with "fabricated benign surrogates." When a refusal occurs, the adaptive retry (Eq. 8) revises the prompt to elicit a new response from the target model itself. The model conditions on responses it did generate, just under a revised prompt. This is consistent with our autoregressive conditioning claim.
>
> > If you disable history curation entirely...what is the resulting SFR?
>
> **A**: Thanks for this question. The condition you describe is already tested: the "Adaptive Recovery" ablation disables retry (Eq. 8), preserving refusals in history as-is. SFR still reaches ~74.3%, substantially exceeding single-turn baselines. We will revise ablation labels to clarify this distinction.
>
> > Table 1 caption says "HarmBench" but Table 3 uses JailbreakBench...not just LLaMA-3-8B
>
> **A**: Thanks for this suggestion. To our knowledge, prior multi-turn baselines (ActorAttack, X-Teaming) report exclusively on HarmBench without JailbreakBench evaluation. Our Table 3 already extends beyond this standard. That said, we agree broader coverage strengthens the paper and have conducted additional JailbreakBench experiments on GPT-4o and Gemini-2.0-Flash:
> | Method | LLaMA-3-8B | GPT-4o | Gemini-2.0-Flash | Avg. Token Cost |
> |---|---:|---:|---:|---:|
> | ActorAttack | 37.5% | 46% | 42% | 15K |
> | X-Teaming | 85.5% | 94% | 86% | 37K |
> | STAR (Ours) | 94.0% | 96% | 100% | 29K |
>
> > Have you conducted any calibration analysis or agreement check of the GPT-4o judge?
>
> **A**: Thanks for this suggestion. We adopt the identical GPT-4o judge and rubric as X-Teaming, which reports 84.5% agreement with the HarmBench classifier (93.2% human agreement).
>
> **3.Recent work**
> > it would be great to add existing LLM multi-turn survey such as https://arxiv.org/abs/2501.09959 and https://arxiv.org/abs/2504.04717.
>
> **A**: Thanks for sharing and we will included it in our related work section in the final version.

---

> > ### Author Rebuttal · Reviewer_LHv1 · 2026-04-03
> >
> > Thank you for your response, my questions have been resolved.

---

> > > ### Author Response · Authors · 2026-04-08
> > >
> > > Dear reviewer,
> > >
> > > Thank you once again for your thorough and constructive feedback, and for acknowledging that our responses have resolved your questions. Thank you!
> > >
> > > Best regards,
> > >
> > > The Authors

---

### Official Review · Reviewer_p7hX · 2026-03-24

**Soundness:** 3
**Presentation:** 3
**Significance:** 2
**Originality:** 3
**Overall Recommendation:** 4
**Confidence:** 3

**Summary:**

This paper introduces STAR, a state-oriented diagnostic framework for multi-turn jailbreaking and safety analysis. It treats dialogue history as a state transition operator and enables controlled analysis of safety ehavior along interaction trajectories. Experimental results demonstrate that STAR is valid acorss different models and outperform baseline methods.

**Compliance With Llm Reviewing Policy:**

Affirmed.

**Key Questions For Authors:**

- It would be valuable to see how the proposed framework performs on more recent frontier models
- Could the authors further elaborate on potential defense implications or mitigation strategies?

**Limitations:**

Yes

**Strengths And Weaknesses:**

Strengths:
- The proposed STAR is technically sound and provides valuable insights to the community as a diagnostic framework
- The empirical experimental results are detailed, and demonstrate strong performance of the proposed STAR compared to baseline methods
- The refusal direction dynamics and latent state trajectories are very clear in supporting the effectiveness of the method
- The ablation analysis is detailed

Weaknesses:
- The models used in the experiments appear somewhat outdated and may not fully reflect the capabilities of state-of-the-art systems as of 2026. This raises concerns about the representativeness and current relevance of the empirical findings.
- Minor: The paper introduces a relatively heavy set of notations, some of which may not be strictly necessary and could affect readability.

---

> ### Author Rebuttal · Authors · 2026-03-30
>
> We appreciate your insightful discussion. We are encouraged that you finds our work technically sound, our empirical results detailed, and our interpretability analysis very clear in supporting the effectiveness of the method. Here we address your detailed comments, which are helping us to revise the paper and chart out future directions.
>
> **1.Limited evaluation on recent frontier models**
>
> > The models used in the experiments appear somewhat outdated and may not fully reflect the capabilities of state-of-the-art systems as of 2026. This raises concerns about the representativeness and current relevance of the empirical findings.
> It would be valuable to see how the proposed framework performs on more recent frontier models
>
> **A**: Thank you for this suggestion. We have conducted additional experiments on two recent frontier models using HarmBench dataset: the open-weight Qwen3.5-27B and the closed-source GPT-5.4-mini:
>
> | Model        | STAR SFR (%) |
> |:-------------|:------------:|
> | Qwen3.5-27B  |     100      |
> | GPT-5.4-mini |     100      |
>
> On Qwen3.5-27B, all six categories are eventually compromised, but the average number of turns varies considerably, from 2.50 for Illegal Activities to 4.88 for Harassment & Bullying, with Misinformation also requiring a longer trajectory (3.75 turns). GPT-5.4-mini exhibits a similar pattern of category-dependent efficiency. Cybercrime & Intrusion is reached most quickly (2.33 turns), while Misinformation & Disinformation (4.38), Illegal Activities (3.88), and Chemical & Biological (3.78) require more sustained interaction. These results suggest that the residual safety barrier is not uniform across harm categories, certain categories remain farther from the compliance region and require longer state evolution before failure occurs.
>
> **2.Notation and readability**
>
> > The paper introduces a relatively heavy set of notations, some of which may not be strictly necessary and could affect readability.
>
> **A**: Thank you for this suggestion. We will carefully review each notation for necessity and streamline the presentation in the revised manuscript.
>
> **3.Defense implications and mitigation strategies**
>
> > Could the authors further elaborate on potential defense implications or mitigation strategies?
>
> **A**: Thanks for this question. Our analysis suggests a concrete defense direction: **joint anomaly detection** over two complementary signals rather than monitoring refusal activation alone.
>
> A single refusal-projection threshold is insufficient because benign multi-turn conversations (e.g., technical discussions, creative writing) naturally operate in low refusal-activation regions. The distinguishing signal our framework reveals is the co-occurrence of two conditions: (i) suppressed refusal-related features (Section 5.4.1) and (ii) progressively strengthening role-conditioned representational coupling. As shown in Figure 6, the alignment score between prompt context and generated representations rises from 0.147 at Turn 1 to 0.282 at Turn 3, reflecting increasing role dominance. The mutual information analysis (Appendix A, Figures 6–7) further shows that named entities introduced during role initialization produce localized peaks in MI with the compliance outcome, confirming that role anchoring tokens act as discrete representational triggers that shift the model from policy-conditioned to role-conditioned generation. This co-occurrence pattern is absent in benign interactions and provides a more reliable detection criterion.
>
> Our ablation results support early interception as a practical strategy. Removing role initialization alone causes a 17.8% SFR drop (Section 5.2), confirming that state initialization is a primary vulnerability. Together, these findings suggest a two-stage defense pipeline: (1) detecting role-conditioned initialization signals (professionally framed personas combined with semantically softened queries) in the first one to two turns, and (2) monitoring the joint trajectory of refusal suppression and role-coupling escalation across subsequent turns. Thanks for this question once again, we will expand this discussion in the final version.

---

> > ### Author Rebuttal · Reviewer_p7hX · 2026-04-04
> >
> > Thanks for the clarification. I will keep my positive score.

---

> > > ### Author Response · Authors · 2026-04-08
> > >
> > > Dear reviewer,
> > >
> > > Thank you once again for your constructive feedback and for acknowledging our rebuttal. We greatly appreciate your positive assessment and thoughtful suggestions. Thank you!
> > >
> > > Best regards,
> > >
> > > The Authors

---

### Decision · Program_Chairs · 2026-04-30

**Decision:**

Accept (regular)

**Comment:**

The reviewers assigned final scores of 2, 4, 4, and 4 to this paper. Although Reviewer Lhv1 (score: 2) acknowledged that their questions had been fully addressed in the authors’ response, they did not revise their score. After reviewing the complete rebuttal and comparing it with the comments of other reviewers, I believe that most of Reviewer Lhv1’s concerns have been sufficiently resolved. As the Area Chair, I recommend acceptance of this paper.